# Inflammasomes in Cardiovascular Diseases: Current Knowledge and Future Perspectives

**DOI:** 10.3390/ijms26125439

**Published:** 2025-06-06

**Authors:** Mario Caldarelli, Laura Franza, Sebastiano Cutrupi, Martina Menegolo, Francesco Franceschi, Antonio Gasbarrini, Giovanni Gambassi, Rossella Cianci

**Affiliations:** 1Department of Medical and Surgical Sciences, Catholic University of Sacred Heart, 00168 Rome, Italy; mario.caldarelli01@icatt.it (M.C.); cutrupisebastiano@gmail.com (S.C.); menegolomartina@hotmail.it (M.M.); antonio.gasbarrini@unicatt.it (A.G.); giovanni.gambassi@unicatt.it (G.G.); 2Fondazione Policlinico Universitario A. Gemelli, Istituto di Ricerca e Cura a Carattere Scientifico (IRCCS), 00168 Rome, Italy; francesco.franceschi@unicatt.it; 3Emergency, Anesthesiological and Reanimation Sciences Department, Fondazione Policlinico Universitario A. Gemelli-IRCCS of Rome, 00168 Rome, Italy; cliodnaghfranza@gmail.com; 4Emergency Department, Azienda Ospedaliero-Universitaria di Modena, 41125 Modena, Italy

**Keywords:** inflammasomes, NLRP3, cardiovascular diseases, atherosclerosis, heart failure, pericarditis

## Abstract

Chronic inflammation is an important contributor to the development of cardiovascular disorders, and inflammasomes, especially the NOD-like receptor protein 3 (NLRP3), are emerging as crucial mediators in this context. Inflammasomes are activated through receptor-mediated danger signals, such as cholesterol crystals and cellular damage products, thereby stimulating the secretion of pro-inflammatory cytokines, which sustains inflammation. This mechanism drives atherosclerosis (via plaque formation and destabilization), heart failure (via fibrotic remodeling), and pericarditis (via exacerbation of pericardial inflammation). Therapeutic approaches seek to block inflammasome activation or their pro-inflammatory pathways. Colchicine, interleukin-1 inhibitors (anakinra, canakinumab), and Sodium-Glucose Transport Protein 2 (SGLT2) inhibitors have a positive impact on cardiovascular inflammation. Various new compounds, such as MCC950, have been described as novel specific inhibitors of NLRP3. Further studies are needed to validate the effectiveness and safety of these treatments. Further elucidating the role of inflammasomes in cardiovascular disease could open the way to achieving more effective therapies, allowing for better management of high-risk cardiovascular patients.

## 1. Introduction

Chronic inflammation is one of the main mechanisms involved in the onset and progression of cardiovascular diseases. This sustained inflammatory process is driven by the inflammasomes NOD-, LRR-, and pyrin domain-containing protein 3 (NLRP3) [1]. These intracellular multiprotein complexes stand as molecular platforms to sense pathogen-associated molecular patterns (PAMPs) and damage-associated molecular patterns (DAMPs) and induce inflammatory cascade through the caspase-1-dependent maturation of IL-1β and IL-18 cytokines [2,3].

Inflammasome activation is also linked to oxidative stress, a hallmark of many chronic diseases, such as atherosclerosis and heart failure [4]. In fact, mitochondrial reactive oxygen species (ROS) act as triggers and amplifiers for NLRP3 inflammasome activation by liberating thioredoxin-interacting protein (TXNIP) from its inhibitory complex, thus allowing TXNIP to bind directly to NLRP3 [2]. The interplay between oxidation and inflammation sustains chronic inflammation, endothelial dysfunction, and maladaptive tissue remodeling. It follows that any therapeutic strategy aimed at controlling both oxidative stress and inflammation would more comprehensively tackle the pathological mechanisms of cardiovascular diseases as it would more directly target the inflammasomes [5]. The involvement of ROS, TXNIP, and lysosomal rupture in inflammasome activation is depicted in Figure 1.

In this narrative review, we report the results of a systematic literature search across PubMed/MEDLINE, Scopus, Web of Science, and Embase electronic databases. The search strategy was geared toward studies on inflammasomes and cardiovascular diseases. To this end, we used a variety of Medical Subject Headings (MeSHs) and free-text keywords, including “inflammasome”, “NLRP3”, “cardiovascular diseases”, “atherosclerosis”, “heart failure”, “pericarditis”, “pyroptosis”, “IL-1β”, “IL-18”, and “targeted therapy”.

We considered only publications from 2015 to capture the most relevant advances that occurred in the last decade. Publications had to be peer-reviewed in the English language, and they could comprise original research (both clinical and experimental), systematic reviews, meta-analyses, and randomized controlled trials. We excluded case reports, conference abstracts, editorial comments, and any other published material that was not peer-reviewed.

Other articles included were screened manually. Articles were selected to highlight recent discoveries and/or novel therapeutic strategies to illustrate the most up-to-date and comprehensive overview of the role of inflammasomes in the pathophysiology of cardiovascular and renal disorders. To gauge their future therapeutic potential, this review also integrates information about experimental compounds currently under investigation.

## 2. Inflammasomes

Inflammasomes are multiprotein complexes expressed predominantly in macrophages and dendritic cells, key components of the innate immune system. Acting as intracellular sensors, they detect danger signals, such as PAMPs and DAMPs, via pattern recognition receptors (PRRs), thereby regulating the production of cytokines—including pro-inflammatory and anti-inflammatory types—in response to injury or infection, with a significant role in immune homeostasis [6,7].

Based on their homologous protein domains, PRRs are grouped into five major families: Toll-like receptors (TLRs), absent in melanoma 2 (AIM2)-like receptors, nucleotide-binding oligomerization domain-like receptors (NLRs), retinoic acid-inducible gene-I (RIG-I)-like receptors, and C-type lectin receptors [1].

Most known inflammasome structures comprise a receptor protein originating from NLR, an apoptosis-associated speck-like protein containing a caspase recruitment domain (CARD), an adapter protein apoptosis-associated speck-like protein containing a CARD (ASC), and an effector protein [8].

The NLRP3 inflammasome is a multimeric protein complex composed of three elements, namely NLRP3, ASC, and caspase-1. The NLRP3 receptor protein contains a central nucleotide-binding NACHT domain, leucine-rich repeats (LRRs), and a pyrin domain (PYD). The adaptor protein ASC is composed of two interaction domains, PYD and CARD. Upon inflammasome assembly, the NLRP3-PYD domain interacts with the ASC-PYD domain and the ASC-CARD domain interacts with the effector protein CARD domain, thus joining the receptor and effector protein, with ASC serving as a necessary link. The stimulation of the receptor protein through this interaction can lead to the activation and cleavage of caspase-1. Upon activation, caspase-1 cleaves pro-inflammatory cytokines, such as IL-1β and IL-18, leading to their maturation and downstream pro-inflammatory signaling [8].

AIM2 contains a PYD and a C-terminal interferon-inducible p200 (HIN200) domain. It is an interferon-induced sensor that recognizes parts of double-stranded DNA (dsDNA) [9]. The interaction of dsDNA triggers AIM2 polymerization, resulting in the development of an inflammasome composed of AIM2, ASC, and caspase-1. This sensor is essential for the detection of dsDNA from a bacterial or viral origin, and it is considered an important player in host defense [10]. AIM2 is triggered by stress, radiation, the replication of DNA, and oxidative damage to nuclear DNA (nDNA) in several cell types [11].

NLRP3 inflammasome activation in the heart is tightly regulated and consists of two steps: priming and triggering [5]. While monocytes efficiently express inflammasome components and secrete IL-1β, cardiomyocytes show a low expression of NLRP3 and caspase-1 and predominantly activate IL-18 [12]. The expression of NLRP3 is significantly increased after ischemia–reperfusion injury, between 3 and 24 h, with the highest levels after 1–3 days, further aggravating the inflammatory response and resulting injury [13]

The active NLRP3 inflammasome stimulates the recruitment and activation of caspase-1, which in turn processes pro-IL-1β and pro-IL-18 to their active forms and cleavages gasdermin D (GSDMD) [14]. Pores formed by GSDMD in the cell membrane result in pyroptosis, a form of pro-inflammatory cell death [15]. During this process, IL-1β, IL-18, and alarm molecules like high-mobility group proteins 1 (HMGB1) are released, further facilitating the amplification of danger signals from dying or damaged cells and stimulating the recruitment of immune cells, especially neutrophils [16]. Oligomeric inflammasome particles can also be phagocytosed by macrophages, further promoting inflammation [17].

Myeloproliferative neoplasms (MPNs) with JAK2 mutation represent a paradigmatic disease used to study the role of chronic inflammation in the cardiovascular system, as complications of this disease include thrombosis, stroke, and myocardial infarction [18]. In this case, the production of pathologic leukocytes and platelets releases pro-inflammatory cytokines, leading to a chronic inflammatory state.

Reactive oxygen species (ROS) production is highly integrated into the NLRP3 inflammasome. There are mainly two events in the activation of NLRP3: that is, the formation of mitochondrial ROS and the dissociation of thioredoxin-interacting protein (TXNIP) from thioredoxins. Thioredoxins are known to be antioxidant proteins that balance the reduction of oxidation in a cell [19]. Under normal conditions, TXNIP functions as a regulatory protein that binds to thioredoxins. However, at elevated levels of ROS, TXNIP detaches from thioredoxins and binds directly to NLRP3, triggering its activation and initiating an inflammatory cascade.

ROS are key endogenous danger signals regulating the NLRP3 inflammasome. Mitochondria are one of the main sources of ROS, particularly in oxidative stress or mitochondrial damage. The accumulation of ROS causes TXNIP to dissociate from the inhibitory complex with thioredoxin. Upon release, TXNIP binds to NLRP3 directly and leads to its activation [19].

In addition, calcium release through ER stress can aggravate mitochondrial dysfunction, facilitating ROS generation, followed by a sustained process of inflammasome activation. Lysosomal breakage and cathepsin B release are also involved. These processes are critical in pathophysiological conditions, such as atherosclerosis, with evidence of an increased expression of ROS and TXNIP in unstable plaques [20].

Indeed, mitochondrial ROS and TXNIP do not act as independent regulators of each other; rather, they work in succession: mitochondrial reactive oxygen species cause the dissociation of TXNIP from thioredoxin, followed by the coupling of free TXNIP to NLRP3 and the activation of this receptor.

These molecular events collectively promote inflammasome assembly and lead to the production of pro-inflammatory cytokines such as IL-1β [4,21,22].

Antioxidant proteins, such as thioredoxins, maintain intracellular redox homeostasis. High levels of intracellular ROS may cause Thioredoxin Interacting Protein (TXNIP) to unbind from thioredoxins and bind to NLRP3, causing its activation [4]. Endoplasmic reticulum (ER) stress also promotes the activation of NLRP3 due to TXNIP overexpression and Ca^2+^ release, which may compromise mitochondrial integrity and lead to ROS generation [23]. The lysosome is also involved in NLRP3 activity. Lysosomal membranes can rupture, leading to cathepsin B release into the cytosol and subsequent NLRP3 activation [24]. Additionally, several microRNAs (miR-223, miR-7, and miR-1929-3p) negatively regulate NLRP3 inflammasome activity at the post-transcriptional level [25]. NLRP3 activation at a genetic level is controlled through negative regulatory, post-transcriptional, and post-translational mechanisms [26]. At the post-translational level, the deubiquitinating enzyme Breast Cancer gene (BRCA)1/BRCA2-containing complex subunit 3 promotes NLRP3 activation [27]. Conversely, NLRP3 ubiquitination is antagonized by F-box and leucine-rich repeat proteins, Tripartite Motif-Containing Protein 31, and Membrane-Associated Ring-CH-Type Finger 7, resulting in the attenuation of inflammasome activation [28]. Phosphorylation by Jun N-terminal kinase and protein kinase D promotes NLRP3 activation, whereas protein tyrosine phosphatase nonreceptor 22 or protein phosphatase 2A acts after dephosphorylation to activate NLRP3 [29].

A growing body of evidence demonstrates that NLRP3 activation is also post-transcriptionally regulated by long non-coding RNAs [30].

## 3. Inflammasomes and CVDs

In the last few years, the inflammasome has been involved in many cardiovascular diseases (CVDs), including atherosclerosis, heart failure, atrial fibrillation, pericarditis, and even chronic kidney disease when there is cardiovascular involvement [1,22]. These diseases are heterogeneous in their clinical presentation, but they all contain converging mechanisms that stem from inflammasome activation, mostly through the NLRP3-IL-1β axis, mitochondrial distress, oxidative stress, and pyroptosis [19,31]. The systemic link between chronic inflammation and cardiovascular dysfunction is well documented in diseases like heart failure or chronic kidney disease, where circulating levels of inflammasome-related cytokines are often associated with poor prognosis [32].

In the following sections, we will discuss the role of inflammasomes in CVD and provide details about shared downstream mechanisms versus disease-specific pathways.

### 3.1. Atherosclerosis

The NLRP3 inflammasome has recently been considered a critical modulator of cardiovascular pathology, including atherosclerosis (AS) [33]. AS is characterized by the progressive accumulation of fatty plaques in the sub-endothelium, resulting in arterial thickening. Plaques contain fatty acids, cholesterol, calcium, fibrin, and cellular waste products. The NLRP3 inflammasome, stimulated by cholesterol crystals and hypoxic signals, not only initiates inflammation but also increases the instability of atherosclerotic plaques. While the role of cholesterol crystals in activating the NLRP3 inflammasome and promoting IL-1β and IL-18 secretion is well established, uncertainties remain about the threshold of crystal size or density needed to trigger this response. Additionally, most evidence comes from in vitro studies, raising questions about the exact relevance to in vivo plaque dynamics. This inflammatory cascade also enhances plaque instability, making NLRP3 a promising target for therapeutic interventions [34]. Hypoxia in atherosclerotic plaques activates the macrophage transcription of hypoxia-inducible factor 1-alpha (HIF-1α), driving a pro-inflammatory cascade marked by VEGF-A secretion and leukocyte infiltration. This feedback loop exacerbates plaque growth, instability, and inflammation [35,36]. Hypoxia and cholesterol crystals play a key role in the activation of the NLRP3 inflammasome in atherosclerotic plaques, as evidenced by studies demonstrating the interaction between HIF-1α/VEGF-A (vascular endothelial growth factor A) and membranous damage mechanisms [37]. HIF-1α can enhance inflammation by repressing the expression of peroxisome proliferator-activated receptor gamma (PPAR-γ), an important anti-inflammatory gene [38]. PPAR-γ encodes the anti-inflammatory transcription factor PPAR-γ, which suppresses pro-inflammatory cytokines and gene expressions, including IL-6, IL-1β, TNF, matrix metalloproteinase-9, and nitric oxide synthase 2, whereas it activates anti-inflammatory genes, such as the IL-10 and heme oxygenase 1 genes [39]. The hypoxia-mediated activation of HIF-1α in macrophages inhibits PPAR-γ and, hence, promotes plaque inflammation by reactivating pro-inflammatory pathways and shutting down anti-inflammatory ones [40].

In addition, the GSDMD-N-terminus, a product of the NLRP3-mediated cleavage of GSDMD, triggers pyroptosis in endothelial cells [41].Concurrently, VEGF-A, a critical cytokine involved in angiogenesis, promotes lesion growth, but it may also act as a pro-inflammatory mediator with destabilizing and rupture-promoting effects [42]. HMGB1 activates the nuclear factor kappa-light-chain-enhancer of activated B cells (NF-κB) pathway to induce the secretion of inflammatory mediators and to indirectly impair endothelial barrier function. Pyroptosis by NLRP3-mediated cell death has also been suggested to drive endothelial cell death [43]. Furthermore, the aggregation or crystallization of cholesterol within the endo-lysosomal system is also capable of damaging lysosomal membranes and releasing cathepsins, thus contributing to NLRP3 inflammasome activity in AS [44]. Cholesterol accumulation in macrophages due to a loss of ABCA1 and ABCG1 cholesterol efflux transporters triggers NLRP3 inflammasome activation and atherosclerosis exacerbation but not lysosomal damage [34]. New studies show that the lysosomal depletion of cholesterol, especially in Niemann-Pick C1, triggers NLRP3 activation [45]. This pathway is linked to the internalization of oxidized LDL or cholesterol crystals and worsens with defective cholesterol efflux. Moreover, checkpoints for detecting endoplasmic reticulum (ER) cholesterol levels and ER stress responses are potentially implicated in the activation of the NLRP3 inflammasome [34]. While it is plausible that large cholesterol crystals lead to NLRP3 inflammasome activation via membrane damage in advanced atherosclerotic plaques, the underlying mechanisms and their relevance remain to be elucidated [46].

Numerous studies have linked NLRP3 inflammasome activation to elevated levels of pro-inflammatory cytokines, particularly IL-1β and IL-18, which correlate with the severity of atherosclerotic disease [47]. In particular, research has consistently indicated a link between NLRP3 activity and elevated levels of IL-1β and IL-18, correlating with the severity of conditions such as coronary artery disease [48,49]. For example, Shateri et al. reported the co-elevation of NLRP3 and IL-1β in patients with coronary artery diseases (CADs), although this study had limitations due to its small sample size and failure to explore confounding factors. In some additional studies, elevated NLRP3 mRNA expression was also witnessed in unstable atherosclerotic plaques from patients who underwent resection by carotid endarterectomy, as opposed to that in stable plaques [50].

Recent studies in atherosclerosis have shed light on some less obvious triggers of inflammasome activation, beyond the usual suspects like cholesterol crystals and low oxygen levels. One interesting player in this game is octanal, a volatile aldehyde that comes from lipid oxidation.

A study by Orecchioni et al. showed that the activation of Olfactory receptor 2 (OLR2) in vascular macrophages by its ligand, octanal, induces AS through the NLRP3 inflammasome pathway [51]. Octanal, a lipid peroxidation product in plasma and atherosclerotic lesions, aggravated AS when injected, while the transplantation of OLR2−/− bone marrow into low-density lipoprotein receptor knockout mice lessened atherosclerosis and the effect of octanal injection [51]. Octanal was found to increase cAMP levels, calcium fluxes, and ROS generation in macrophages [33]. Octanal in combination with LPS also increased IL-1β and lactate dehydrogenase secretion, a marker of membrane permeability, indicating that octanal might function as a second signal for NLRP3 inflammasome activation. However, the treatment also enhanced the secretion of IL-1α and TNF-α, highlighting more general pro-inflammatory effects [52].

All other inflammasomes remain poorly understood in atherosclerosis, except AIM2, NLRP1, and NLRC4. Borborema et al. observed that patients with atherosclerosis had significantly higher gene expression levels of NLRP1 and NLRC4 compared to healthy controls [53]. Bleda et al. showed that high triglyceride and very low-density lipoprotein (VLDL) levels are related to increased NLRP1 inflammasome activation in human arterial endothelial cells [54]. This increased expression correlates with the degree of endothelial dysfunction [55]. The AIM2 inflammasome has also emerged as a pro-inflammatory entity in atherosclerosis. In atherosclerotic plaques, AIM2 colocalizes with the DNA in foam cells, while a lack of AIM2 diminishes IL-1β and IL-18 levels [56].

In summary, the activation of the NLRP3 inflammasome by cholesterol crystals and hypoxic signals not only initiates inflammation but also enhances the instability of atherosclerotic plaques through IL-1β- and IL-18-mediated mechanisms. All these processes make NLRP3 a potentially valuable therapeutic target in the treatment of CVDs.

### 3.2. Heart Failure

The inflammatory mechanisms that induce instability in atherosclerotic plaques—especially NLRP3 activation, IL-1β signaling, and endothelial dysfunction—are also central to myocardial remodeling and heart failure. As demonstrated by Toldo et al., inflammasome activation contributes to fibrotic and hypertrophic changes in cardiomyocytes, particularly during ischemic injury [5]. Heart failure (HF) represents a major global health challenge, partly driven by population aging and improved cardiac care, which has prolonged survival in patients with cardiovascular diseases [57]. As of now, around 56 million people globally suffer from HF [58].

There is extensive evidence of an association between HF progression and the overexpression of biologically active molecules. Important mediators include the stimulation of the sympathetic nervous system (SNS), the renin–angiotensin–aldosterone system (RAAS), and the natriuretic peptide system, each involved in cardiac repair and remodeling [59].

HF can be classified in different ways, based on the time of onset (acute and chronic) [60], its severity in terms of symptoms (NYHA classification) [61], or the ejection fraction [62]. The latter is of particular interest, as it allows us to identify a subset of patients with HF who have a preserved ejection fraction but are still symptomatic: this group is defined by an ejection fraction of over 50% [63]. In this group, the role of the immune system in determining the onset of the disease is particularly relevant [64]. Current evidence suggests that myocardial injury activates cells to induce short-term myocardial adaptation via the activation of innate and adaptive immune systems [32].

NLRP3, AIM2, and NLRC4 inflammasomes play a crucial role in HF via the activation of caspase-1 and enabling pathways such as pyroptosis and pathological remodeling. AIM2 is significantly overexpressed in diabetic animal models, and its knockdown has been shown to reduce myocardial damage [65]. NLRP3 is involved in pressure overload-induced cardiac hypertrophy by mechanisms such as muscle LIM protein S-nitrosylation to activate IL-1β signaling and amplify myocardial dysfunction [66]. One of the main changes brought about by chronic inflammation that promote pathological changes in cardiomyocytes and non-myocytes is fibrosis [67]. NLRP3 inflammasome activation leads to the production of fibrogenic IL-1β and IL-18, which accelerate the evolution of fibrosis [68].

NLRP3 activation induced via calcium-/calmodulin-dependent protein kinase IIδ consequent to pressure overload also worsens myocardial fibrosis and heart failure [66].

Pyroptosis, resulting from NLRP3 inflammasome activation, may worsen myocardial dysfunction, as it has been associated with doxorubicin-induced dilated cardiomyopathy and pathological heart remodeling [69]. Pyroptosis could also be associated with increased cardiac fibrosis through fibroblast activation and inflammatory cell recruitment [69].

Several studies have suggested that modulating inflammasome activity can reduce myocardial hypertrophy and fibrosis [70], and it has been observed that people with a lower ejection fraction have a greater presence of cardiac neutrophils and a higher activity range of NLRP3 [71]. NLRP3 can modulate the activity of cardiac macrophages, which are heavily involved in the process of adverse heart remodeling in HF. In knockout murine models, cardiac remodeling due to macrophage accumulation appears to be reduced [72].

Modulating inflammasome activity offers promising avenues for reducing cardiac fibrosis and inflammation, as evidenced by preclinical models. Both NLRP3 and NLRP1 emerge as key players: while NLRP3 inhibition mitigates fibrotic and inflammatory responses, NLRP1 targeting suppresses pathways like mitogen-activated protein kinase (MAPK), NF-κB, and Tumor Growth Factor-β (TGF-β)/Smad, demonstrating multifaceted potential for therapeutic interventions [72,73]

A role of inflammasomes in HF can also be indirectly envisaged. After myocardial infarction, IL-1β is upregulated and promotes adverse cardiac remodeling while also negatively affecting the β-adrenergic pathway and promoting myocyte dysfunction [74]. IL-18 has similar effects, especially affecting the activation of the β-adrenergic receptors, leading to heart injury [75].

### 3.3. Atrial Fibrillation

In addition to ventricular remodeling, activation of the inflammasome has a role in atrial structure and function. Dobrev et al. reported that in atrial cardiomyocytes, excessive NLRP3 activity increases arrhythmogenesis, calcium dysregulation, and fibrotic remodeling, predisposing patients to atrial fibrillation (AF) [76].

The NLRP3 inflammasome has been indicated as a key player in multiple types of atrial cardiomyopathy and AF [76]. Its functional expression in the atria has significant clinical implications for AF, meaning it is involved in the pathogenesis of AF.

Studies in patients with mitral valve disease and AF revealed an upregulated NLRP3 inflammasome and pro-inflammatory M1-type macrophages [77]. Similarly, in patients with end-stage ischemic heart failure, those with AF showed an increased number of infiltrating macrophages as well as activated caspase-1. Infiltrating macrophages were the prime suspects of NLRP3 inflammasome activity [78].

Macrophages with a pro-inflammatory phenotype play an important role in atrial inflammation and thrombogenesis in AF [79]. Although it was first characterized in immune cells, the NLRP3 inflammasome is also expressed in endothelial and epithelial cells, cardiomyocytes, and fibroblasts. Excessive inflammasome activity in atrial cardiomyocytes and fibroblasts has been shown in recent studies to be both necessary and sufficient to trigger AF [80].

A new mouse model with fibroblast-specific NLRP3 activation showed left-atrial enlargement and fibrosis, gap junction remodeling, and AF vulnerability [81].

The NLRP3 inflammasome is activated by various stimuli in AF, ultimately leading to IL-1β-driven inflammatory signaling, immune cell recruitment, fibrosis, and electrical instability [82]. Early evidence associates IL-1β with altered myocardial contractility, ion channel activity, and Ca^2+^ handling abnormalities, which aggravate atrial myopathy and post-operative AF [83].

IL-1β can be derived from a range of sources, including cardiomyocytes, macrophages, and epicardial adipose tissue. The pro-arrhythmic activity of IL-18 is less clear, but its levels correlate with action potential duration in atrial cardiomyocytes and will need to be further explored [84].

### 3.4. Chronic Kidney Disease and the NLRP3 Inflammasome

Not all inflammatory mechanisms are restricted to the heart. Recent studies suggest that the NLRP3 inflammasome may be an important mediator in the cardiorenal axis [85].

Chronic kidney disease (CKD) is both a progressive disease and a significant risk factor for CVD. The NLRP3 inflammasome is increasingly being implicated as an important mediator that connects renal inflammation with cardiovascular complications [19].

In CKD, the NLRP3 inflammasome is upregulated in renal tubular epithelial cells, podocytes, and infiltrating immune cells. The activation of the NLRP3 inflammasome through the pro-caspase-1-dependent generation of IL-1β and IL-18 and pyroptosis enhances renal fibrosis, glomerular injury, and proteinuria. In patients with advanced CKD, elevated circulating levels of inflammasome-related cytokines have been associated with aortic stiffening, left ventricular hypertrophy, and accelerated atherosclerosis [86].

Mitochondrial dysfunction and oxidative stress are general hallmarks of CKD and can trigger thioredoxin-interacting protein (TXNIP) release, which directly activates the NLRP3 inflammasome. In addition, uremic toxins and endoplasmic stress lead to inflammasome activation, further promoting local and systemic inflammation [85].

Chronic sterile inflammation in CKDs leads to vascular calcification, endothelial dysfunction, and the infiltration of immune cells into cardiovascular tissues, making inflammasome activation an attractive target for intervention in both kidney and cardiac diseases [87].

## 4. Special Focus: Pericarditis and Still’s Disease

CKD truly represents a systemic inflammatory pathology driven by the inflammasome, whereas the pericardium appears to be one site of inflammasome activity. Recurrent or idiopathic pericarditis, as well as systemic inflammatory diseases, like Still’s disease and numerous other arterial disorders, exemplifies the auto-inflammatory role of the NLRP3-IL-1β axis in cardiovascular pathology that is not associated with atherosclerosis. These diseases illustrate how inflammasomes can prompt sterile inflammatory events different from the usual cardiovascular risk factors [88].

Pericarditis, the most common disease of the pericardium, is characterized by inflammation of the pericardial sac. It can be classified into acute, subacute, chronic, and recurrent types [89].

In developed countries, viral infections are the leading cause of pericarditis, in the same way that tuberculosis is the leading cause in developing countries. Among the most common non-infectious causes are autoimmune diseases, cancer, and post-cardiac injury syndrome [88].

In the Western world, the incidence of acute pericarditis in the general population is 27.7 cases per 100,000 persons per year, while the standardized incidence rate of hospital admissions for pericarditis is 3.32 cases per 100,000 person-years [90].

Recurrent idiopathic pericarditis is the result of two distinct mechanisms: autoimmunity and auto-inflammatory processes [91]. Autoimmunity is typical of rheumatologic disorders like systemic lupus erythematosus, while auto-inflammatory processes are strongly connected with NLRP3 inflammasome activation and IL-1β release [92]. Genetic studies have shown that Mediterranean fever (MEFV)-1 gene variations, which also occur in auto-inflammatory conditions such as familial Mediterranean fever (FMF), contribute to susceptibility to recurrent pericarditis. The sharing of MEFV-1 as a common gene between FMF and recurrent pericarditis underscores the importance of the NLRP3 inflammasome in both conditions. Colchicine, which is effective in FMF, demonstrates comparable results in pericarditis, solidifying the link between pathogenesis and therapeutic approaches [91,93].

Pericarditis frequently co-occurs in Still’s disease (~90%), with genetic links such as CARD8 polymorphism exacerbating inflammation [94,95]. Still’s disease is a rare inflammatory disorder that includes both systemic juvenile idiopathic arthritis (sJIA) in children and adult-onset Still’s disease (AOSD) in adults [96]. AOSD is characterized by recurrent episodes of high peak fever (>39–40 °C), a transient pink-salmon rash, and arthralgia or polyarthritis. Laboratory findings typically include an elevated neutrophilia and total white blood cell (WBC) count, reflecting significant systemic inflammation [97]. The diagnosis of AOSD is primarily clinical, after the exclusion of infections, hematologic diseases such as lymphoma, cancer, and other autoimmune or auto-inflammatory diseases. Examples of such conditions include vasculitis, periodic fever syndromes, and the recently described VEXAS (vacuoles, E1 enzyme, X-linked, auto-inflammatory, somatic) syndrome [98].

The pathophysiology of AOSD is not fully understood. Like most autoimmune and auto-inflammatory diseases, it is thought to involve an interplay between genetic predisposition and environmental factors [99]. This complex interaction likely contributes to the dysregulated immune response observed in the disease. Joung et al. observed a strong association between AOSD and the human leukocyte antigen (HLA)-DRB112 and HLA-DRB115 alleles. However, they also reported a negative association with HLA-DR1 and HLA-DRB1*04, suggesting that certain HLA genotypes may influence susceptibility to AOSD [100]. The molecular mechanisms underlying inflammasome activation involve a wide range of upstream signals, including ROS, ER stress, mitochondrial dysfunction, and cholesterol crystals, which converge on TXNIP and lead to NLRP3 activation. The full pathway is illustrated in Figure 2.

Because many clinical disorders appear to be associated with inflammasome activation, it is hypothesized that there are overlapping molecular mechanisms common to these diseases. Table 1 brings together the principal stimuli, sensors, downstream molecular players, and pathophysiological effects linked with NLRP3 inflammasome activation in cardiovascular and renal diseases.

## 5. New Pharmacology Strategies

Sodium–glucose co-transporter 2 (SGLT2) inhibitors, originally intended for glycemic control in type 2 diabetes, are now among one of the drug classes recommended by the US Food and Drug Administration (FDA) with robust cardiovascular benefits. Apart from glucose reduction, these drugs have anti-inflammatory, antioxidation, and anti-remodeling properties [101].

Mechanistically, SGLT2 inhibitors seem to decrease inflammation through various ways. They act as oxidative stress inhibitors, mitochondrial function regulators, and NLRP3 inflammasome inhibitors. Data indicate that SGLT2 inhibitors decrease mitochondrial ROS generation, one of the most relevant activators of NLRP3. They may also directly inhibit NLRP3 assembly by inducing ketogenesis and transitioning cellular energy metabolism towards the production of β-hydroxybutyrate, which has been demonstrated to inhibit NLRP3 activation in macrophages [102].

In addition, SGLT2 inhibitors enhance autophagic flux and ionic homeostasis, including Na^+^ and Ca^2+^ levels, inside cells, which helps to preserve mitochondrial integrity and mitigate inflammatory response, along with restoring mitochondrial defects. Other data also indicate that SGLT2 inhibitors may be involved in decreasing pro-inflammatory cytokines, IL-1β, and IL-6 in cardiac tissue, as well as in systemic circulation [103].

These pleiotropic properties may be responsible for the effects on cardiovascular mortality, hospitalization for HF, and adverse cardiac remodeling, which has been observed especially in HF patients. Therefore, SGLT2 inhibitors can be considered modulators of inflammasome-induced cardiac inflammation.

The anti-inflammatory, antioxidant, and anti-remodeling effects of SGLT-2 are now recognized to extend beyond HFrEF and HFpEF and may also benefit other forms of cardiac remodeling.

Inhibiting NLRP3 offers several advantages over directly targeting IL-1 activity, and orally administered NLRP3 inhibitors are now available.

Anakinra (an IL-1 receptor antagonist active in both IL-1β and IL-1α) has been approved for various inflammatory conditions, but its cardioprotective effects are not conclusive. Among patients with non-ST-segment-elevation myocardial infarction (NSTEMI) in the MRC-ILA Heart Study, anakinra treatment led to a transient downregulation of C-reactive protein (CRP) and IL-6 levels relative to the placebo, but it was associated with higher rates of major adverse cardiac events (MACEs) in one year [104]. On the other hand, in the VCU-ART3 trial, patients with ST-segment-elevation myocardial infarction (STEMI) given anakinra within 12 h of symptom onset showed reduced CRP levels and a lower incidence of new HF- and heart-related hospitalizations [105]. Future studies should address whether dosage or timing adjustments might enhance therapeutic benefit while reducing possible risks.

The CANTOS study was the largest trial of cytokine inhibition, involving 10,061 patients, which demonstrated the important role of inflammation in CVDs [106]. In patients with prior myocardial infarction and elevated CRP, the IL-1β inhibitor, canakinumab, reduced cardiovascular event risk and the levels of IL-6 and CRP but did not affect LDL cholesterol and showed a small increase in fatal infections.

Colchicine is another drug to consider as an anti-inflammatory therapy for cardiovascular prevention according to the 2021 ESC guidelines [107]. Colchicine blocks microtubule activity in rapidly proliferating cells and specifically inhibits IL-1β release through NLRP3 without affecting other inflammasomes [108]. It has been approved for the secondary prevention of CVDs at a low dose (0.5 mg daily), and in June 2023, the Food and Drug Administration (FDA) approved it to reduce the risk of myocardial infarction, stroke, and cardiovascular death in patients with atherosclerotic disease or multiple risk factors [109].

In the low-dose colchicine trial, low-dose colchicine reduced the risk of acute coronary syndrome and related events by 67% in patients with stable coronary artery disease [110]. Its effectiveness in HF remains controversial; in some experimental HF models, colchicine resulted in a reduction in myocardial apoptosis and inflammation, preserving LV function and improving survival [111].

The use of colchicine has also been considered in the secondary prevention of CVDs. However, moving from limited clinical trials to widespread application will require further evidence of the risk–benefit balance in different patient subgroups.

Despite the development of numerous NLRP3 antagonists, none have been approved for use in humans [112]. The direct targeting of NLRP3 offers increased specificity as it does not interfere with the activation of other NOD family members. This suggests that the risk of opportunistic infections observed with anti-IL-1 biologics may be reduced with NLRP3 inhibitors. In addition, a shorter half-life may be beneficial in situations where prolonged immunosuppression is a risk [112].

In acute pericarditis, the first-line drugs are aspirin or non-steroidal anti-inflammatory drugs (NSAIDs) [113]. Treatment should be continued for 2 weeks and discontinued if the pain resolves [114].

Low, weight-adjusted doses of colchicine are recommended to improve the response to medical therapy and to help reduce the incidence of recurrences [115]. Colchicine has been proposed for standard adjunct treatment with NSAIDs [116].

While in cases of incomplete response to aspirin/NSAIDs and colchicine, corticosteroids are often used, these agents should be given as a triple therapy combined with low-to-moderate dose aspirin/NSAIDs and colchicine rather than replacing these agents, [117].

A proportion of patients remain resistant to standard therapy, even at the maximum tolerated doses.

In such situations, symptoms can last or recur when the dose is brought down, or in the case of stopping the treatment. In patients unresponsive to conventional therapy, several biopharmaceuticals have been evaluated, and IL-1 inhibitors have shown promise [118].

Therapies like Rilonacept and Goflikicept, which act against IL-1α and IL-1β, prevent the recurrence of pericarditis by inhibiting inflammation and allowing patients to discontinue corticosteroids without recurrence [119].

A new treatment proposal involves hydroxychloroquine (HCQ), which appears to possess steroid-sparing action with a favorable cost–benefit profile [120]. In a pilot observational study of 30 patients, HCQ did not significantly lower the risk of recurrence; however, it led to a prolonged median time to flare-free survival by four weeks and to a reduction in the hazard ratio for flares. In addition, 33.3% of HCQ patients were able to reduce their glucocorticoid dose by 50%, as compared to none in the control group, and overall, glucocorticoid dosing was lower (43.5% vs. −4.5%, *p* < 0.001). The treatment was well tolerated, with no discontinuations [120].

The chronic activation of the NLRP3 inflammasome can lead to chronic inflammation, most commonly due to TLR4 signaling. These pathways have inspired novel therapeutic approaches targeting TLR4, such as TAK-242 (resatorvid), and have been investigated in disorders like osteoarthritis [121].

MCC950, as a direct inhibitor of NLRP3, represents a step up from drugs such as colchicine and canakinumab that act downstream. The potential of MCC950, which has been demonstrated in preclinical studies to specifically block NLRP3, offers a targeted alternative to current anti-inflammatory therapies that act downstream, such as IL-1 inhibitors. However, the transition from preclinical models to clinical practice poses significant challenges, including potential side effects and bioavailability. MCC950 acts on the NACHT domain of NLRP3 and blocks vital steps for the activation of the inflammasome, providing benefits for myocardial function and attenuating heart failure-related hypertrophy, fibrosis, and inflammation [122].

MCC950 ameliorated cardiac metabolism, pulmonary artery pressure, and ventricular arrhythmias. While these findings are promising, clinical trials have not been conducted. Wang et al. explored the role of the selective NLRP3 inhibitor MCC950 in HF in obese mice to elucidate the associated metabolic pathways. The study showed that MCC950 significantly reduced pressure overload-induced cardiac hypertrophy and fibrosis. It also significantly decreased cardiac inflammation following transverse aortic constriction surgery and enhanced the infiltration of M2 macrophages in cardiac tissues [123].

Oridonin inhibits NLRP3 inflammasome assembly by forming a covalent bond with cysteine (Cys)279 in the NACHT domain of NLRP3, effectively blocking its activation [124].

OLT1177 does interact directly with the NLRP3 inflammasome to lower its ATPase activity and block ASC oligomerization, thereby preventing NLRP3 assembly but leaving the NLRC4 or AIM2 inflammasomes unaffected [125].

Preclinical in vivo research by Toldo et al. showed that OLT1177 preserves myocardial function after ischemia–reperfusion injury in mice [126]. When given at the time of reperfusion, it reduced the infarct size in a dose-dependent manner, and it continued to do this even when given 60 min post-reperfusion.

Disulfiram has been reported to decrease atherosclerotic lesions and inflammatory responses in hyperlipidemic preclinical in vivo models [127].

Disulfiram was also found to induce autophagy in a variety of cell types, including macrophages, smooth muscle cells, endothelial cells, hepatocytes, and tissue samples from atherosclerotic plaques [128].

On the NF-κB inhibitor side, BAY 11-7082 is a compound that inhibits NF-κB by targeting the phosphorylation of IκBα [129]. By preventing IκBα phosphorylation and the subsequent translocation of NF-κB to the nucleus in response to TNF-α, BAY 11-7082 effectively suppresses NLRP3 inflammasome activation [130]. After oxygen–glucose deprivation followed by reoxygenation, BAY 11-7082 reduces the levels of both the NLRP3 inflammasome and cleaved caspase-1 protein, highlighting its potential role in mitigating inflammasome-related inflammation and injury under these conditions [131]. Kim et al. hypothesized that BAY 11-7082 could mitigate myocardial injury following ischemia/reperfusion (I/R) [132]. In a rat myocardial I/R injury model, BAY effectively inhibited NF-κB p65 activation, reducing infarct size, preserving systolic function, and decreasing fibrosis and apoptosis. These results suggest that BAY may help mitigate myocardial injury associated with I/R [132].

Other strategies are also under investigation to block the NLRP3 inflammasome pathway and include targeting ASC, NEK7, and gasdermin D. However, the primary focus has been on the direct inhibition of NLRP3, starting with the most prominent NLRP3-targeting molecule, originally known as CRID3 and now referred to as MCC950 [133].

In Table 2, there is a summary of the current pharmacological strategies available to target NLRP3, directly and indirectly.

## 6. Conclusions

An increasing body of research is documenting the role played by inflammasomes in driving harmful processes such as cytokine release, fibrogenesis, endothelial dysfunction, and pyroptosis in conditions like atherosclerosis, heart failure, pericarditis, atrial fibrillation, and chronic kidney disease. Their activation, especially NLRP3, seems to be involved in mediating the sterile inflammation observed in several cardiovascular and cardiorenal pathologies.

While most of the current therapies target downstream mediators like IL-1β, the search for specific NLRP3 inhibitors is still ongoing. It remains critical to how individual diseases mobilize distinct patterns of activation and how signaling through the inflammasomes intersects with oxidative stress and metabolic dysfunction to establish crosstalk between organs (such as on the heart–kidney axis). Further research in this field may lead to an era of precision medicine with more integrated approaches for truly high-risk patients.

## Figures and Tables

**Figure 1 ijms-26-05439-f001:**
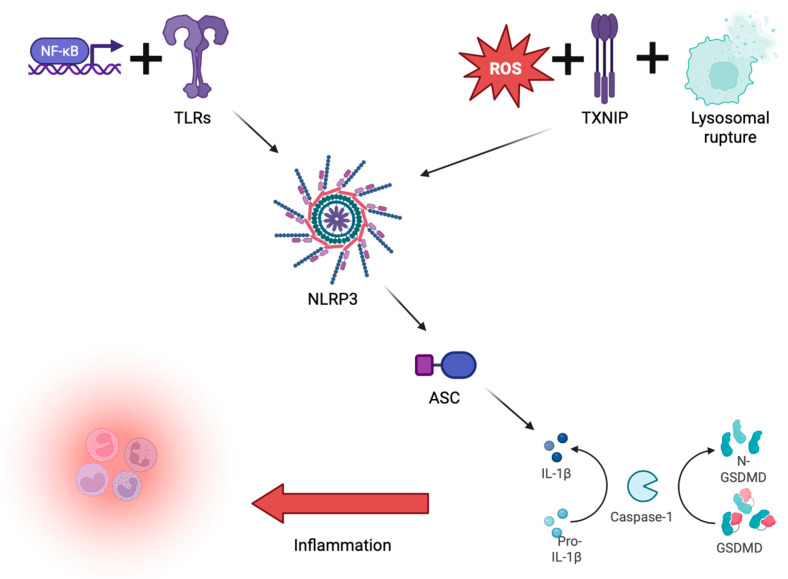
Priming signals, such as the activation of Toll-like receptors (TLRs) through NF-κB, result in the upregulation of the expression of both NLRP3 and pro-IL-1β. Reactive oxygen species (ROS), thioredoxin-interacting protein (TXNIP) dissociation, and lysosomal rupture are activation signals that trigger the oligomerization of NLRP3. This is conducive to recruiting the signaling adaptor protein ASC and pro-caspase-1, producing the active inflammasome complex. Then, caspase-1 cleaves pro-IL-1β into mature IL-1β and gasdermin D (GSDMD), generating N-terminal GSDMD (N-GSDMD), which forms membrane pores and induces pyroptosis. Tissue inflammation ensues, culminating in the release of inflammatory cytokines and increased recruitment of immune cells. Figure Created in BioRender. Mario Caldarelli. (2025) https://app.biorender.com/illustrations/6834a90177126cc127ecb968?slideId=2778b30d-7e7a-49eb-89c4-5b654ea7eda0 (accessed on 26 May 2025).

**Figure 2 ijms-26-05439-f002:**
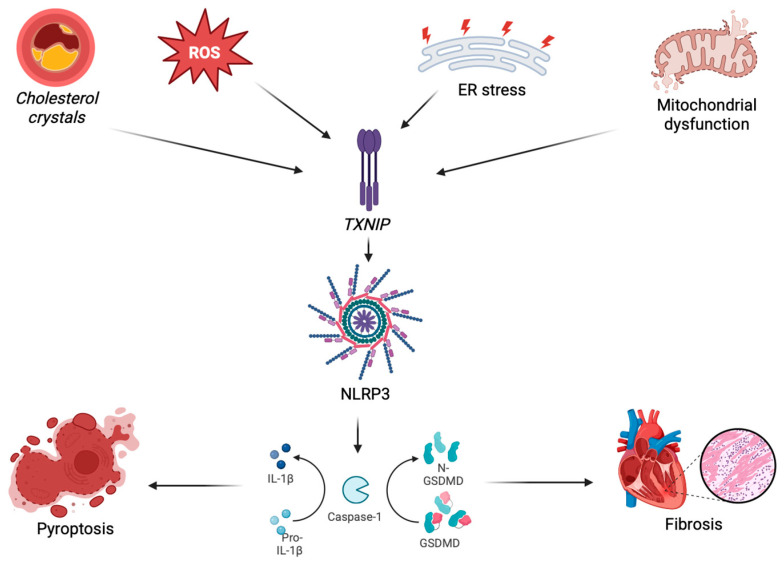
The activation of the NLRP3 inflammasome may lead to pathogenic processes in cardiovascular and renal disorders. Though various cellular stressors can cause injury to endothelial cells, impaired function, and thus the activation of thioredoxin-interacting protein (TXNIP), upon activation, the NLRP3 localizes ASC and pro-caspase-1 for the latter to cleave pro-IL-1β into IL-1β and gasdermin D (GSDMD) into its N-terminal form (N-GSDMD). These processes lead to pyroptosis and favor cardiac fibrosis in the onset of cardiovascular and cardiorenal diseases. Figure Created in BioRender. Mario Caldarelli (2025) https://app.biorender.com/illustrations/68349cf16b644c3b97338e6b?slideId=2be5472e-8701-4acd-9302-b3a741c620c5 (accessed on 26 May 2025).

**Table 1 ijms-26-05439-t001:** Major stimuli and signaling pathways leading to NLRP3 inflammasome activation and their pathological outcomes in cardiovascular and renal diseases.

Trigger	Sensor/Pathway	Key Molecules	Downstream Effect	Associated Conditions	References
Cholesterol crystals	Lysosomal rupture	NLRP3, ASC, caspase-1	IL-1β, IL-18 release, pyroptosis	Atherosclerosis	[44,46]
ROS/oxidative stress	ROS–TXNIP–NLRP3	TXNIP, NLRP3, caspase-1	Inflammation, fibrosis	Heart failure, CKD	[19,20]
Mitochondrial dysfunction	mtROS, mtDNA	NLRP3, GSDMD	Pyroptosis, cytokine secretion	AF, HF, CKD	[19]
ER stress	Ca^2+^ signaling	NLRP3	IL-1β production, cell damage	CKD, myocardial injury	[23,24]
Uric acid/ATP	P2X7 receptor activation	K^+^ efflux, NLRP3	Inflammasome priming/activation	Pericarditis, CKD	[19,88]
Mechanical stretch	Ion channels, ROS	NLRP3, NF-κB	Fibrosis, arrhythmias	Atrial fibrillation	[76,79,81]

**Table 2 ijms-26-05439-t002:** Pharmacological strategies currently available for the direct and indirect targeting of NLRP3.

Molecule	Target	Type of Study	References
Anti-SGLT-2	Sodium–glucose cotransporter-2	Clinical	[101]
Anakinra, Canakinumab	IL-1	Clinical	[104,105]
Colchicine	Interacts with tubulin and thereby inhibits microtubule-dependent functions in rapidly proliferating cells	Preclinical in vitro	[108]
TAK-242 (resatorvid)	TLR4	Preclinical	[121]
MCC950	NACHT domain of NLRP3.	Review of preclinical studies	[123]
Oridonin	NACHT domain of NLRP3	Preclinical	[124]
OLT1177	Inhibits ASC oligomerization	Preclinical	[126]
Disulfiram	Causes autophagy in a variety of macrophages, smooth muscle cells, endothelial cells, hepatocytes, and tissue samples from atherosclerotic plaques	Preclinical	[127]
BAY 11-7082	Inhibits NF-κB	Preclinical	[132]

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
