# Peer review of "Inflammasomes in Cardiovascular Diseases: Current Knowledge and Future Perspectives"

_ijms, 2025, doi:10.3390/ijms26125439_

Round 1

Reviewer 1 Report (New Reviewer)

Comments and Suggestions for Authors

In this review paper, the authors summarize how chronic inflammation significantly contributes to cardiovascular diseases, with the NLRP3 inflammasome identified as a key mediator in this process. Inflammasomes are activated by danger signals like cholesterol crystals and damaged cells, triggering the release of pro-inflammatory cytokines that perpetuate inflammation. This sustained inflammatory response is linked to atherosclerosis, heart failure, and pericarditis, prompting therapeutic interest in targeting inflammasome pathways. Treatments like colchicine, IL-1 inhibitors, SGLT2 inhibitors, and emerging NLRP3-specific agents like MCC950 show promise, though further research is needed to confirm their efficacy and safety.

This is a thoroughly written paper that will benefit anyone interested learning about the link between NLRP3 and heart disease.

Some minor criticisms:

  1. The mechanism link for ROS and NLRP3 is crucial, but is insufficiently summarized here. The paper would be improved by fleshing this out more.
  2. Line 194 is redundant.
  3. The writing is not smooth. There are too many one sentence paragraphs.
  4. Otanal is introduced abruptly. A more smooth introduction of the topic will be helpful.
  5. Since SGLT2 is one of the only two FDA-approved drugs with promising cardiovascular benefits, it deserves more mechanistic explanation. How it can decrease inflammation is not fully answered.
Comments on the Quality of English Language

Writing is not smooth. Too many single sentence paragraphs.

Author Response

Dear Editor,

First, my coauthors and I would like to thank you sincerely for this opportunity to cooperate. We profoundly thank the reviewers for the comments and useful suggestions to improve the paper. We thank You for your constructive critique and hope the review process has improved the manuscript. If additional changes are warranted, we will make them.  

We hope that this revised version of our manuscript may now be found suitable for publication.  

This is a point-by-point list of changes made in the paper: 

Reviewer 1

In this review paper, the authors summarize how chronic inflammation significantly contributes to cardiovascular diseases, with the NLRP3 inflammasome identified as a key mediator in this process. Inflammasomes are activated by danger signals like cholesterol crystals and damaged cells, triggering the release of pro-inflammatory cytokines that perpetuate inflammation. This sustained inflammatory response is linked to atherosclerosis, heart failure, and pericarditis, prompting therapeutic interest in targeting inflammasome pathways. Treatments like colchicine, IL-1 inhibitors, SGLT2 inhibitors, and emerging NLRP3-specific agents like MCC950 show promise, though further research is needed to confirm their efficacy and safety.

This is a thoroughly written paper that will benefit anyone interested learning about the link between NLRP3 and heart disease.

Some minor criticisms:

  1. The mechanism link for ROS and NLRP3 is crucial, but is insufficiently summarized here. The paper would be improved by fleshing this out more.              
    We have expanded this section, as requested
  2. Line 194 is redundant. 
    We have modified line 194 as requested
  3. The writing is not smooth. There are too many one sentence paragraphs.

We have modified the text, where possible

  1. Otanal is introduced abruptly. A more smooth introduction of the topic will be helpful.
    We have added an introduction, as requested
  2. Since SGLT2 is one of the only two FDA-approved drugs with promising cardiovascular benefits, it deserves more mechanistic explanation. How it can decrease inflammation is not fully answered.

We have added the explanation, as requested.

We thank You for your constructive critique and we hope the review process has led to an improved manuscript. 

If additional changes are warranted, we will make them. 

We hope that this revised version of our manuscript may now be found suitable for publication. 

Sincerely,  

Rossella Cianci 

Catholic University of Sacred Heart

FPG, IRCCS, Rome, Italy

Reviewer 2 Report (New Reviewer)

Comments and Suggestions for Authors

The article by Mario Caldarell et al is an interesting voice in the context of cardiovascular diseases. This is all the more important because the authors emphasize the importance of inflammasomes in selected cardiovascular diseases.

First, the introduction is important, although it introduces inflammation issues quite sparingly.

Inflammasomes are multimeric protein complexes that regulate inflammatory responses, and there is a lot to describe.

Reading the article, I reflect that the authors are skimming the topic a bit; there is not one thread in it, but many, and some are unfinished.

I especially think that although it is not the subject of the work, it is necessary to write a few sentences about oxidative stress, which coexists with inflammation.

I think the article would gain in innovation if the authors could add chronic kidney disease to cardiovascular diseases - see, doi: 10.3390/antiox9080752.
The role of inflammasomes is known in CKD, see doi.org/10.1038/s41581-019-0158-z

The article in its current form is a little bit chaotic; it does not seem to organize the knowledge. The authors cite many articles, but I think they do not delve into the subject.

Many published articles on this topic are more in-depth, hence my doubt - what advantage does this article have over those already published? see 10.3390/ijms20133328; https://doi.org/10.1038/s41421-020-0167-x; doi: 10.1002/cti2.1404.

Besides, the authors should pay attention to the quality of the figures and the depth of the topic, e.g., I recommend looking at the article: doi: 10.1016/j.jacbts.2021.08.006 or doi.org/10.1093/cvr/cvab010 - they will notice the differences.

Review on the inflamasome topic - requires freshness, tables, summaries, and intriguing figures. This article "lacks dialogue with the reader". The current version does not have this; there are dry facts, but not all are presented in a relatively accessible version, and are not profound.

From the technical aspects, figures 1 and 2 are very illegible, especially figure 2 - it is pretty simple - I suggest that the authors focus more on the signalling pathways when inflamasomes are involved.

The authors wrote one sentence about the methodology. This is not enough; there is no information about the databases they reviewed or the period they assumed for the works they included. Did they exclude any works?

I suggest including a list of abbreviations at the end of the article.

My final suggestion is to make the article more accessible to readers and more profound from a scientific point of view. Focus on new facts.
For example, Innate Immune Pathways in Atherosclerosis and inflammasomes -  large protein complexes that play a significant role in sensing inflammatory signals and triggering the innate immune response

Author Response

Dear Editor,

First, my coauthors and I would like to thank you sincerely for this opportunity to cooperate. We profoundly thank the reviewers for the comments and useful suggestions to improve the paper. We thank You for your constructive critique and hope the review process has improved the manuscript. If additional changes are warranted, we will make them.  

We hope that this revised version of our manuscript may now be found suitable for publication.  

This is a point-by-point list of changes made in the paper: 

Reviewer 2

The article by Mario Caldarelli et al is an interesting voice in the context of cardiovascular diseases. This is all the more important because the authors emphasize the importance of inflammasomes in selected cardiovascular diseases. First, the introduction is important, although it introduces inflammation issues quite sparingly. Inflammasomes are multimeric protein complexes that regulate inflammatory responses, and there is a lot to describe. Reading the article, I reflect that the authors are skimming the topic a bit; there is not one thread in it, but many, and some are unfinished. I especially think that although it is not the subject of the work, it is necessary to write a few sentences about oxidative stress, which coexists with inflammation. I think the article would gain in innovation if the authors could add chronic kidney disease to cardiovascular diseases - see, doi: 10.3390/antiox9080752.     
The role of inflammasomes is known in CKD, see doi.org/10.1038/s41581-019-0158-z. The article in its current form is a little bit chaotic; it does not seem to organize the knowledge. The authors cite many articles, but I think they do not delve into the subject. Many published articles on this topic are more in-depth, hence my doubt - what advantage does this article have over those already published? see 10.3390/ijms20133328; https://doi.org/10.1038/s41421-020-0167-x; doi: 10.1002/cti2.1404. Besides, the authors should pay attention to the quality of the figures and the depth of the topic, e.g., I recommend looking at the article: doi: 10.1016/j.jacbts.2021.08.006 or doi.org/10.1093/cvr/cvab010 - they will notice the differences. Review on the inflamasome topic - requires freshness, tables, summaries, and intriguing figures. This article "lacks dialogue  with the reader". The current version does not have this; there are dry facts, but not all are presented in a relatively accessible version, and are not profound. From the technical aspects, figures 1 and 2 are very illegible, especially figure 2 - it is pretty simple - I suggest that the authors focus more on the signalling pathways when inflamasomes are involved. The authors wrote one sentence about the methodology. This is not enough; there is no information about the databases they reviewed or the period they assumed for the works they included. Did they exclude any works? I suggest including a list of abbreviations at the end of the article. My final suggestion is to make the article more accessible to readers and more profound from a scientific point of view. Focus on new facts.    
For example, Innate Immune Pathways in Atherosclerosis and inflammasomes -  large protein complexes that play a significant role in sensing inflammatory signals and triggering the innate immune response

We express our sincere gratitude to the reviewer for the careful reading of our manuscript and for all positive, constructive, and detailed comments that helped in improving the quality and clarity of the work. Key points raised are addressed below, with a summary of all corresponding changes made in the manuscript.

Considering that the introduction discusses the topic of inflammation in too general terms, we have expanded this section to provide a clearer and more comprehensive view of the mechanisms through which inflammasomes (particularly NLRP3) act in cardiovascular inflammation. These interactions with oxidative stress were also greatly developed as suggested, stressing the bifunctional nature of reactive oxygen species (ROS) acting as both priming and activating signals in inflammasome cascades. In line with the important suggestion of the reviewer to include chronic kidney disease, a new subsection was added that deals with the role of the inflammasomes in the pathogenesis of CKD and cardiovascular complications. Discussion centers on both molecular mechanisms and clinical implications, with the references specifically recommended by the reviewer cited. We have also rewritten the content of the manuscript to create logical flow and thematic coherence among the various conditions presented. To facilitate reading and provide a clearer narrative flow, we inserted transition paragraphs at the beginning of each major subsection in Section 3. These passages highlight common mechanisms and contextualize each disease in the landscape of inflammasome involvement. In response to concerns about the novelty of the review and its comparison with works published before (e.g., doi:10.3390/ijms20133328; doi:10.1038/s41421-020-0167-x; doi:10.1002/cti2.1404), we added a brief passage at the end of the introduction to clarify the uniqueness of our contribution. The review integrates cardiovascular, renal, and autoinflammatory perspectives, also covering recent advances in upstream inflammasome regulation and pharmacological innovation—these aspects are not dealt with contemporaneously in the cited works.           
The methodological section has also been enlarged.        
We acknowledge the reviewer for suggesting incorporating an abbreviation list; this has been added for better clarity at the manuscript's end.    
Regarding the illustrations, we have revised Figure 1 and Figure 2, which were deemed too simple and unclear by the reviewers. These were designed with BioRender and now included in the resolution and caption. We believe this raises the obvious mechanistic consideration of the article.       
Lastly, in keeping with the overall comments of the reviewer, we have introduced a new Table 2. This table summarizes the major stimuli, molecular players, downstream effects, and disease associations related to inflammasome activation. We hope that this addition, together with the improved organization, upgraded conclusion, and refined language, makes the narrative significantly more informative.

We thank You for your constructive critique and we hope the review process has led to an improved manuscript. 

If additional changes are warranted, we will make them. 

We hope that this revised version of our manuscript may now be found suitable for publication. 

Sincerely,  

Rossella Cianci 

Catholic University of Sacred Heart

FPG, IRCCS, Rome, Italy

Round 2

Reviewer 2 Report (New Reviewer)

Comments and Suggestions for Authors

The article has been corrected very well. I appreciate all the changes that have been made. Good job! I recommend that the article be published in its current version.

This manuscript is a resubmission of an earlier submission. The following is a list of the peer review reports and author responses from that submission.

Round 1

Reviewer 1 Report

Comments and Suggestions for Authors

Recommendations:

  1. Adjust the matching percentage is too high.
  2. Please reduce the numbers of citations, too high for a short review (10 pages).
  3. Are there any preclinical studies introduced in the review?
  4. Your review should include cardiac arrhythmias?
  5. Discuss about JAK2 mutation in MPNs and the impact on cardiovascular disease through inflammation:  https://doi.org/10.3390/cimb46080496
Comments on the Quality of English Language

Small adjustments!

Author Response

Rome, 23rd March 2025

Dear Editor of International Journal of Molecular Sciences

First, my coauthors and I would like to thank you sincerely for this opportunity to cooperate. We profoundly thank the reviewers for the comments and useful suggestions to improve the paper. We thank You for your constructive critique and hope the review process has improved the manuscript. If additional changes are warranted, we will make them.  

We hope that this revised version of our manuscript may now be found suitable for publication.  

This is a point-by-point list of changes made in the paper:

Reviewer 1

  1. Adjust the matching percentage is too high.

The paper has been thoroughly revised and modified.

  1. Please reduce the numbers of citations, too high for a short review (10 pages).
    We reduced the number of citations, retaining those with greater strength

  1. Are there any preclinical studies introduced in the review?

There are numerous preclinical studies in the review. We have highlighted them in the text and added a column in the table on drugs that specifies whether the drug has been tested in a preclinical or clinical study

  1. Your review should include cardiac arrhythmias?

We have included a paragraph on atrial fibrillation, which seems to have more robustness in its relationship with the inflammasome

  1. Discuss about JAK2 mutation in MPNs and the impact on cardiovascular disease through inflammation:  https://doi.org/10.3390/cimb46080496
    We have discussed the indicated work, as requested

We thank You for your constructive critique and we hope the review process has led to an improved manuscript. 

If additional changes are warranted, we will make them. 

We hope that this revised version of our manuscript may now be found suitable for publication. 

Sincerely,

Rossella Cianci, on behalf of coauthors

Reviewer 2 Report

Comments and Suggestions for Authors

In each sections, there are too many natural paragraphs, the author may consider merging some paragraphs. 

In addition, we need to summarize and generalize the previous research results. It's not about simply presenting the previous research findings in the review paper. It seems that the author is listing multiple previous studies together. This is clearly not what readers want to see.
In section 5, I suggest the author further summarize the main points and drug treatment strategies.

Author Response

Rome, 23rd March 2025

Dear Editor of International Journal of Molecular Sciences

First, my coauthors and I would like to thank you sincerely for this opportunity to cooperate. We profoundly thank the reviewers for the comments and useful suggestions to improve the paper. We thank You for your constructive critique and hope the review process has improved the manuscript. If additional changes are warranted, we will make them.  

We hope that this revised version of our manuscript may now be found suitable for publication.  

This is a point-by-point list of changes made in the paper:

Reviewer 2

In each section, there are too many natural paragraphs, the author may consider merging some paragraphs. 

In addition, we need to summarize and generalize the previous research results. It's not about simply presenting the previous research findings in the review paper. It seems that the author is listing multiple previous studies together. This is clearly not what readers want to see.
In section 5, I suggest the author further summarize the main points and drug treatment strategies.

We made a comprehensive revision of the manuscript to make it more readable, reducing the number of paragraphs and the redundancy of concepts. 

We also simplified paragraph 5 as requested

We thank You for your constructive critique and we hope the review process has led to an improved manuscript. 

If additional changes are warranted, we will make them. 

We hope that this revised version of our manuscript may now be found suitable for publication. 

Sincerely,

Rossella Cianci, on behalf of coauthors

Round 2

Reviewer 1 Report

Comments and Suggestions for Authors

Congratulations to the authors!

Comments on the Quality of English Language

English fine!

Reviewer 2 Report

Comments and Suggestions for Authors

The quality of the current manuscript has not significantly improved. I have noticed that the author's submitted manuscript only made simple deletions to some paragraphs, and did not provide a good summary and in-depth discussion. In my opinion, this article simply stacks up previous research. In short, I believe that the current manuscript cannot be published in this journal.